# Citrus Bergamia and Cynara Cardunculus Reduce Serum Uric Acid in Individuals with Non-Alcoholic Fatty Liver Disease

**DOI:** 10.3390/medicina58121728

**Published:** 2022-11-26

**Authors:** Yvelise Ferro, Samantha Maurotti, Elisa Mazza, Roberta Pujia, Angela Sciacqua, Vincenzo Musolino, Vincenzo Mollace, Arturo Pujia, Tiziana Montalcini

**Affiliations:** 1Department of Medical and Surgical Science, University Magna Grecia, 88100 Catanzaro, Italy; 2Department of Clinical and Experimental Medicine, University Magna Grecia, 88100 Catanzaro, Italy; 3IRC-FSH-Pharmaceutical Biology-Department of Health Sciences, University Magna Grecia, 88100 Catanzaro, Italy; 4Department of Health Science, University Magna Grecia, 88100 Catanzaro, Italy; 5Research Center for the Prevention and Treatment of Metabolic Diseases, University Magna Grecia, 88100 Catanzaro, Italy

**Keywords:** nutraceuticals, uric acid, flavonoids, liver steatosis, hyperuricaemia

## Abstract

*Background and Objectives:* Hyperuricemia and liver steatosis are risk factors for cardiovascular diseases and mortality. The use of natural compounds could be a safe and effective alternative to drugs for the treatment of fatty liver and hyperuricemia. Polyphenolic fraction of Citrus Bergamia in association with the extract of Cynara Cardunculus, as nutraceutical, is able to reduce body weight, hepatic steatosis and markers of oxidative stress. Then, we performed a secondary analysis of a double-blind placebo-controlled trial to examine the effects of this nutraceutical on serum uric acid levels in adults with fatty liver. *Materials and Methods:* The study included 94 individuals with hepatic steatosis. For six weeks, the intervention group was given a nutraceutical (300 mg/day) comprising a Bergamot polyphenol fraction and Cynara Cardunculus extract. The control group received a daily pill of placebo. Serum uric acid, lipids, glucose and anthropometric parameters were assessed at baseline and after 6 weeks. *Results:* We found a greater reduction in serum uric acid in the participants taking the nutraceutical rather than placebo (−0.1 ± 0.7 mg/dL vs. 0.3 ± 0.7 mg/dL, *p* = 0.004), and especially in those with moderate/severe hepatic steatosis also after adjustment for confounding variables. In addition, we analysed the two groups according to tertiles of uric acid concentration. Among participants taking the nutraceutical, we found in those with the highest baseline serum uric acid (>5.4 mg/dL) the greater reduction compared to the lowest baseline uric acid (−7.8% vs. +4.9%; adjusted *p* = 0.04). The stepwise multivariable analysis confirmed the association between the absolute serum uric acid change and nutraceutical treatment (B = −0.43; *p* = 0.004). *Conclusions:* A nutraceutical containing bioactive components from bergamot and wild cardoon reduced serum uric acid during 6 weeks in adults with fatty liver. Future investigations are needed to evaluate the efficacy of this nutraceutical in the treatment of hyperuricaemia.

## 1. Introduction

Uric acid (UA) is an endogenous molecule and the end result of purine metabolism in humans [1,2]. The liver produces UA, which is eliminated via the kidneys (65–75%) and intestines (25–35%) [1]. Serum levels of UA [SUA] are known to increase with age and males have higher SUA levels than females due to gender-factors [1,3,4,5]. Many studies suggest a strong link between SUA and Non-Alcoholic Fatty Liver Disease (NAFLD) [6,7,8], showing that SUA is a risk factor for the development of fatty liver disease [9,10] and progression to Non-Alcoholic Steatohepatitis (NASH) [11,12,13]. Moreover, patients with NAFLD have higher SUA levels compared to individuals without hepatic steatosis [8,9,10]. In addition, high SUA levels are associated with cardiovascular diseases (CVD) [14,15,16,17] and all-cause mortality. However, it is still controversial whether UA is a causal risk factor for CVD [14]. Urate-lowering drugs are agents used to treat hyperuricaemia, capable of modulating the activity of key enzymes responsible for the metabolism and excretion of urates such as xanthine oxidoreductase (XO), urate transporter 1 [URAT1/SLC22A12] e glucose transporter 9 (GLUT9) [18]. Although these medications are well tolerated and safe, they can induce even serious adverse reactions, especially in the elderly and in subjects with renal dysfunction [19,20]. Using natural compounds extracted from fruits and vegetables could so be a safe and effective alternative to treat metabolic diseases such as NAFLD and hyperuricemia [21,22,23]. Several previous studies showed that many bioactive components of Citrus Bergamia (i.e., naringin, neoesperidin, neoeriocitrin, brutieridin, and melitidin) and Cynara Cardunculus (i.e., cynaropicrin, luteolin and chlorogenic acid) can reduce the concentration of SUA, Citrus fruits, artichoke, or their extracts, appear to have effects on both fatty liver and SUA levels [18,22,24,25,26,27,28].

Furthermore, several clinical, preclinical and cellular studies have already been conducted to evaluate the effects of antioxidants from Bergamot alone on hepatic steatosis [29,30,31]. A meta-analysis provides evidence from randomized trials to support the use of artichoke leaf extract as a hepatoprotective agent in patients with NAFLD [32]. A recent evidence has showed that the polyphenolic fraction of bergamot (BPF) in association with the extract of Cynara Cardunculus (CyC), as nutraceutical, can reduce fatty liver content and markers of oxidative stress [33,34]. Based on the previous studies, it is conceivable that a combination of these two nutraceuticals could have effects on SUA concentration.

For this reason, we performed a secondary analysis of a clinical trial to examine the effects of the nutraceutical with BPF and CyC on SUA levels in NAFLD subjects.

## 2. Materials and Methods

### 2.1. Study Design

This is a secondary analysis of a randomized, double-blinded, controlled trial (RCT) that took place at the Clinical Nutrition Unit of the “Mater Domini” Azienda University Hospital in Catanzaro (Italy), during the period from 11 February to 24 June 2019. The trial protocol and the main results were previously published elsewhere [33]. The study was performed in accordance with the Declaration of Helsinki’s guidelines. The protocol of the study was approved by the Local Ethic Committee (N. 219/2018/CE, approved 24 September 2018) and registered in the ISRCTN Registry with Reg. N. ISRCTN12833814. The clinical trial was financed by Italian Ministry of University and Research (MIUR, Nutramed Project, PON 03PE000_78_1, Rome, Italy). All participants were given and signed an informed consent.

### 2.2. Population and Randomization

For the RCT, we enrolled adults who were invited via newspaper advertisements to be screened for hepatic steatosis using transient elastography (TE). We performed the diagnosis of NAFLD on the value of controlled attenuation parameter (CAP) >216 dB/m evaluated by TE as previously described [33]. For the study, we enrolled a population of all genders with hepatic steatosis between the ages of 30 and 75. We excluded patients with NAFLD due to other causes, as previously described [33]. We were also excluded subjects were taking nutraceuticals, dietary supplements, and functional foods to treat hepatic steatosis, and subjects with allergies to nutraceutical components, or maize, and subjects affected by diabetes or with serum triglycerides levels higher 250 mg/dL [33].

We randomized one hundred and two adults with NAFLD in a 1:1 ratio to receive a nutraceutical with BPF and CyC formulated in combination or a placebo for 12 weeks [33]. Subjects were randomized using an Excel random number generator (Microsoft, Seattle, WA, USA) to receive one capsule daily of nutraceutical containing 150 mg of BPF, 150 mg of CyC plus 300 mg of excipients (i.e., polyunsaturated fatty acid, and a mixture of bergamot pulp and albedo derivative) [33]. The control group received one capsule daily of the placebo. Placebo pill contains maltodextrin and the same excipients as the nutraceutical [33].

For this secondary analysis, we only included subjects who had completed the first part of RCT (6 weeks) and had completed data on all parameters. A total of ninety-four participants were included for this analysis.

### 2.3. Outcomes

The primary endpoint of this analysis was the change of SUA after 6 weeks of intervention in patients with NAFLD. Secondary outcome was the changes of SUA according to gender after 6 weeks of intervention. Detailed definitions of RCT’s primary and secondary endpoints were summarized in the original report [33]. All subjects were given oral and written recommendations to follow a Mediterranean Diet without energy intake restriction. Only overweight/obese subjects received a calorie restriction of 400–500 calorie from their baseline energy intake [33]. The investigators and subjects were unaware of who received the placebo or the nutraceutical.

### 2.4. Preparation of Nutraceutical

The specification sheet with the most relevant active ingredients of BPF and CyC (Bergacyn^®^, provided by Herbal & Antioxidant SRL, Bianco, RC, Italy) is reported in the previous study [33]. Using industrial squeezing and pressing, bergamot juice was extracted from peeled-off fruits. The juice was stripped of its oil component, purified by ultra-filtration, and placed onto polystyrene resin columns that absorbed polyphenol compounds with MWs ranging from 300 to 600 Da (Mitsubishi Chemical Group, Chiyoda, Tokyo, Giappone). A mild KOH solution was used to elute BPF. Furthermore, the neutralization of the phytocomplex was done by cationic resin filtration at an acid pH. To produce a powder, it was vacuum dried and minced to the necessary particle size. When given alone or in combination, the powder was micronized and co-grinded with bergamot albedo fibers. We used HPLC tool to analyze the content of flavonoids and other polyphenols in the powder. Toxicological investigations were also conducted, which showed the lack of harmful chemicals, such as pesticides, heavy metals, sinephrine, and phthalate. Usual microbiological testing revealed no bacteria or mycotoxins. The same method was employed to make CyC extract. Bergamot albedo fibers were micronized and co-ground with plant extracts as excipients for final formulations. HEAD srl provided all materials (Bianco, Calabria, Italy). Then, to create a formulation comprising both extracts, 150 mg of CyC was mixed with 150 mg of BPF powder and encapsulated in pills containing 300 mg of excipients represented by micronized and co-ground albedo fibers with plant extracts (Seris srl, Cuneo, Italy). The final formulation comprised 5% cynaropicrin. For the placebo group, pills containing 600 mg of maltodextrin were produced. All pills were placed in bottles in the manner previously outlined [33]. All operations were carried out in accordance with the European Community Guidelines on dietary supplements. Pharmacokinetic studies and toxicological reports are reported in a previous study [33,34].

### 2.5. Anthropometric Measurements and Cardiovascular Risk Factors Assessment

Using a calibrated digital scale (model Tanita BC-418MA, Manchester, UK) with an accuracy of 0.1 kg, the bodyweight was assessed before breakfast following a 12 h overnight fast, with the participants lightly clothed (the clothing weight was subtracted). A stadiometer (wall-mounted) was used to measure the body height. Body Mass Index (BMI) was determined as follows: weight/height2 (kg/m^2^). Obesity was described as having a BMI ≥ 30 kg/m^2^. Hip and waist circumferences (HC and WC) were determined using a non-stretchable tape over light clothing at the level of the widest diameter around the buttocks and over the undressed abdomen at the narrowest point between the costal margin and the iliac crest, respectively. Then, waist to hip ratio (WHR) was calculated.

From patient interviews and clinical records, we investigated the existence of traditional cardiovascular risk factors (i.e., hyperlipidaemia, smoking, and hypertension) [33]. We measured blood pressure at baseline and after 6 weeks. The following criteria were used to define and exclude diabetes: fasting blood glucose ≥126 mg/dL or antidiabetic treatment [35].

### 2.6. Blood Measurements

After an overnight fast, venous blood was obtained in Vacutainer tubes (Becton & Dickinson, Plymouth, UK) and centrifuged for 4 h. Insulin, serum glucose, creatinine, total cholesterol, triglycerides, high-density lipoprotein cholesterol (HDL-C), ALT, AST, γGT, total bilirubin, and UA were assessed by chemiluminescent immunoassay on COBAS 8000 (Roche, Basel, Switzerland), following the manufacturer’s instructions. Low-density lipoprotein cholesterol (LDL-C) level and homeostatic model assessment (HOMA) index were calculated as previously described [33,36].

### 2.7. Statistical Analysis

Data are reported as mean ± standard deviation (M ± SD) or %. The Kolmogorov–Smirnov normality test was employed to analyse the continuous variables’ distribution.

Chi-square test was performed to analyse the prevalence between treatment groups and SUA tertiles. An independent unpaired samples t test was used to compare the difference between means, also after 6 weeks according to severity of NAFLD assessed at baseline. An ANOVA was performed to compare the means between SUA tertiles with a Fisher’s LSD test as a post hoc analysis.

The general linear model (GLM) was performed to adjust SUA change (absolute and percentage values) for known factors that influence the concentrations of SUA such as age, gender, sartans, calcium channel, β blockers, diuretics, urate-lowering drugs, and antiplatelet agents [1,3,37,38]. In the GLM, we have also added the variables that were different at independent unpaired samples t-test (weight change), and at χ2 test (smoking habit).

Finally, in whole population a stepwise multivariable linear regression analysis was performed to test the association between absolute SUA change and the confounding variables that were different at the independent unpaired samples t-test and χ2. In particular, we assessed the relationship between SUA change and treatment (BPF + CyC or placebo), age, gender, smoking habit, sartans, calcium channel, β blockers, diuretics, urate-lowering drugs, antiplatelet agents [1,3,36,37], and change of weight after 6 weeks.

Significant differences were assumed to be present at *p* < 0.05 (two-tailed). SPSS 22.0 for Windows (IBM Corporation, New York, NY, USA) was used to perform all comparisons.

## 3. Results

We analysed data from ninety-four individuals with NAFLD of the ninety-five subjects who completed the first part of the investigation after 6 weeks of treatment. Seven participants were lost within 6 weeks (n. 3 participants in the BPF + CyC group and n. 4 participants in the placebo group) [33], and one patient was excluded from this analysis because he did not have a baseline SUA value.

Table 1 shows the clinical characteristics of participants according to the treatment. At baseline, the groups were comparable for age, BMI, CAP score and SUA. The smoking prevalence was different between groups, and was higher in the BPF + CyC group than the placebo (28% BPF + CyC group vs. 9% placebo group, *p* = 0.03) (Table 1). No participants in the BPF + CyC group took urate-lowering drugs compared to 2 subjects in the placebo group (*p* = 0.49). None of the participants changed their usual drug therapy during the 6 weeks of treatment.

The changes in the clinical parameters after 6 weeks of treatment period are shown in Table 2. BPF + CyC treatment significantly lowered body weight (−3.1 ± 2.3 kg vs. −2.1 ± 1.8 kg, *p* = 0.03; respectively), and BMI (−3.1 ± 2.3 vs. −2.1 ± 1.8 kg/m^2^, *p* = 0.02; respectively) compared to the placebo (Table 2). The change in the SUA levels was different between groups (−0.1 ± 0.7 mg/dL in BPF + CyC group vs. 0.3 ± 0.7 mg/dL in placebo group, *p* = 0.004) (Table 2). GLM was performed to adjust SUA change for age, gender, smoking habit, and drugs, as well as for weight change, confirming the higher reduction in BPF + CyC group than in placebo group (−0.1 ± 0.1 mg/dL vs. 0.3 ± 0.1 mg/dL, *p* = 0.005; respectively) (Table 2).

The individual reduction of SUA percentage according to treatment after 6 weeks is shown in Figure 1.

Figure 2 shows the SUA change according to the severity of NAFLD at baseline. We found that subjects with moderate and severe fatty liver disease (S2 and S3 grade) taking nutraceutical capsules had a greater reduction in SUA concentration compared to placebo also after adjusting for confounding variables (*p* = 0.007 and *p* = 0.04, respectively for NAFLD grade S2 and S3).

When we analysed the group taking the nutraceutical according to tertiles of uric acid concentration, we found the highest reduction in SUA in who had a greater basal SUA level (III tertile, SUA > 5.4 mg/dL; −4.9% vs. + 5.1 % in the III tertile vs. I tertile, *p* = 0.03, Appendix A). In the III tertile, 81% were male and 19% female (Appendix A). The tertiles were comparable to body weight change (*p* = 0.12, Appendix A). We found again the highest reduction (−7.8%) in SUA in the subgroup with a greater basal SUA levels also after adjustment for all the variables which differed between tertiles (gender, medications, and glucose and body weight all at baseline and creatinine change; *p* = 0.04; Figure 3). In participants who took placebo, we did not find any SUA reduction (*p* = 0.16).

Finally, in the entire population we performed a multivariable analysis. In this analysis the absolute SUA change was associated only with BPF + CyC treatment (B = −0.43; *p* = 0.004) (Table 3).

## 4. Discussion

This secondary analysis of a RCT demonstrated, for the first time, that nutraceutical containing bioactive molecules from Citrus Bergamia and Cynara Cardunculus (BPF and CyC, respectively) is able to produce a significant reduction of SUA levels in patients with NAFLD after 6 weeks of treatment.

After 6 weeks, the BPF + CyC nutraceutical significantly reduced SUA by ~1% compared to the placebo group (Table 2). We found that subjects with moderate/severe fatty liver steatosis taking nutraceutical had a greater reduction in SUA level compared to placebo also after adjusting for confounding variables.

In the group taking the nutraceutical, we found a significant reduction in SUA in those with the highest basal SUA level (>5.4 mg/dL, III tertile; Appendix A and Figure 3). In this study, we unfortunately enrolled less female than men (60% males). Among participants in the III Tertile only 19% were females. We cannot therefore exclude the effect also on women.

These results are remarkably interesting, as subjects with NAFLD are known to have higher SUA levels than controls [8,9,10], and that hyperuricaemia is associated with a greater stage of histological liver injury and NASH [11,12,13]. Indeed, experimental studies have shown that a high UA level activates oxidative stress, inflammation, and the fibrosis pathway, mechanisms by which the hyperuricemia can contribute to organ damage and cardio-metabolic diseases [39,40].

Antihyperuricemic drugs may improve surrogate endpoints of CVD such as blood pressure, endothelial function, carotid intima media thickness and proteinuria [14], while the effects on clinical outcomes still remain unknown. The findings obtained in our analysis remain very interesting as they encourage the development of new formulations of natural molecules with nutraceutical effects for the management of metabolic diseases.

It is known that UA is the end result of purine nucleotide catabolism that can be modulated by numerous factors including diet and animal proteins intake that contribute significantly to the purine pool [1,2]. The intermediate products of this catabolism include xanthine and hypoxanthine. XO enzyme catalyzes the final two steps in the biochemical chain which results in UA formation: hypoxanthine is converted to xanthine, which is ultimately converted to UA [18]. At the glomerulus level, urate is freely filtered, but up to 90% of the filtered urate is reabsorbed. URAT1/SLC22A12 and GLUT9 are the main transporters involved in tubular reabsorption [18].

The results obtained on the reduction of SUA concentration seem in line with some preclinical and clinical studies. In our study, the SUA reduction can be attributed to the synergistic effect of the bioactive components of bergamot and artichoke contained in the nutraceutical.

As shown in previous studies [33,34], this nutraceutical contains naringin, neoesperidin, neoeriocitrin, brutieridin, and melitidin derived from Citrus Bergamia, and cynaropicrin, luteolin and chlorogenic acid derived from Cynara Cardunculus.

It is interesting to note that in a randomized study, the daily consumption of an orange fermented drink (500 mL/day) for 2 weeks significantly reduced the concentration of SUA [−8.9%] in healthy subjects [25].

Another study showed that the intake of lemon juice for 6 weeks reduced the SUA levels in subjects with gout from 0.4–2.8 mg/dL (mean 1.6 mg/dL) and in patients with hyperuricemia from 0.3–2.3 mg/dL (mean 1.3 mg/dL) [41].

Another study also showed a decrease in SUA after taking freshly squeezed pure lemon fruit juice daily at 30 mL/day (equivalent to a lemon a day) for 6 weeks in males with hyperuricemia [42]. The same authors demonstrated that lemon juice also reduced SUA in a mice model with hyperuricemia [42].

Another study with hyperuricaemic mice showed that an orange juice caused a reduction of ~25% in XO and ~39% in SUA compared to the control [43]. Mo SF et al. demonstrated that naringenin, a flavonoid of citrus fruits, at a dose of 100 mg/kg after 3 days of treatment, was able to reduce SUA levels by inhibiting XO again in hyperuricemic mice [44].

Furthermore, it has been shown that other citrus fruit molecules such as naringin and neoesperidin also have a slight inhibitory activity on XO compared to hesperitin [44], and this seems to be due to the molecules’ different chemical structures [26,44].

Cynara Cardunculus also appears to reduce XO activity. Indeed, in a model of adult mice with hypercholesterolemia, Cynara Cardunculus has been shown to reduce the levels of creatinine, SUA and urea [28]. Luteolin and chlorogenic acid also reduced SUA levels by inhibiting the activity of XO [45,46].

Other mechanisms independent of the inhibition of XO could also be involved in the hypouricemic action of citrus fruits [42]. In fact, it has been shown that citrus flavonoids such as nobiletin, hesperetin and naringenin inhibit the activity of the URAT1 transporter [greater than 30%] more strongly than other polyphenols [24]. Naringin and neohesperidin also have mild activity on URAT1 [23]. All these evidences suggest that the nutraceutical thanks to its bioactive components seems to act both by reducing the formation and increasing the excretion of UA, thus determining a decrease in the concentration of SUA.

When comparing participants with NAFLD treated with BPF + CyC to the placebo group, we discovered a statistically significant reduction in body weight (Table 2).

These data do not surprise us, as it appears to be in line with findings from other studies, which show that the intake of polyphenols causes a change in metabolism and prevents weight gain or helps weight loss [33,47]. The mechanisms of these potential effects could be due to stimulation of catabolic pathways in liver and adipose tissue, reduction of inflammation related to obesity, increase of glucose uptake by skeletal muscles and others [47]. However, an adjustment for weight change confirmed our finding (Table 2). When we categorized the population in SUA tertiles, we find the greatest reduction in SUA levels in those with a high basal SUA level (III Tertile) and the tertiles were comparable for body weight change (among tertiles *p* = 0.12). We therefore hypothesize an effect of nutraceutical on SUA that is independent of weight loss and change in eating habits.

This study has some limitations. European guidelines for the management of NAFLD recommend, of course, using ultrasound (US) as first-choice imaging in adults at risk for NAFLD [48]. However, in clinical practice, it is well accepted that US limitations include suboptimal sensitivity and specificity for the detection of mild steatosis less than 25–30% of hepatocytes involved and operator dependence. Furthermore, there are also concerns about the possible consequences of overdiagnosis NAFLD in the severe grade. Therefore, to date, it can be used to obtain only a qualitative and semiquantitative assessment of the extent of the disease. Instead, magnetic resonance imaging, the ideal technique, with the highest sensitivity to detect liver steatosis, to date, it is not recommended as a first-line tool given its cost and limited availability in this clinical setting. CAP appears thus to be a new diagnostic tool for non-invasive assessment and quantification of steatosis. The performance of CAP was evaluated in the study of Sasso et al. [49] taking the histological grade of steatosis as reference. AUROC was equal to 0.91 and 0.95 for the detection of more than 10 and 33% of steatosis, respectively. These data showed CAP could efficiently identify several steatosis grades.

A recent study focusing on a head-to-head comparison between US and TE [50], demonstrated that CAP values obtained with TE were comparable with US diagnosis and closely correlated with the steatosis severity of US detected in high-risk subjects. Moreover, in low-risk patients, CAP seems to be more sensitive in steatosis detection compared to US evaluation. The CAP evaluation by FibroScan has been thus suggested to be used to diagnose and detect, with over the 90% of sensitivity, liver steatosis in recent practice guidelines [51]. The CAP value is used also in obese individuals as showed by De Barros et al. [52] and the performance of TE was evaluated in obese taking the histological grade of steatosis as reference [53].

It was a secondary analysis of an RCT designed to evaluate the effects of this nutraceutical on fatty liver accumulation and/or on the derived liver damage markers. Furthermore, our analysis showed only a small reduction in SUA levels compared to that achieved with dietary changes alone [54] or drug treatments [55]. However, this reduction was obtained with only one capsule daily, but it is conceivable that a greater reduction of SUA levels could be obtained by increasing the daily dose of this nutraceutical [34]. For this reason, the findings obtained must be interpreted with caution and can only generate hypotheses for future investigations.

A second limitation is that we evaluated the effect on SUA after 6 weeks of intervention and in subjects not suffering from hyperuricemia. The intervention in the present study was designed to provide proof of principle and only one dose was tested. It would interest to enroll individuals with hyperuricemia in a future clinical trial with a longer study duration. Another limitation of the investigation is that it did not evaluate the quantity of UA in the urine, and urea. We have not even evaluated the relationship between the serum concentration of the bioactive components of the nutraceutical and SUA reduction.

Furthermore, we did not evaluate how the change in eating habits may have affected the SUA variation after 6 weeks. However, the study has some strengths, such as analysing the whole population, assessing the effects by gender, and adjusting the data obtained for known factors that can influence SUA levels such as age, sex, drug use [37,38] and body weight change [56].

## 5. Conclusions

The nutraceutical containing the polyphenolic fraction of bergamot and the extract of artichoke leaves reduces the blood levels of UA and body weight after 6 weeks of treatment in adults with NAFLD. This nutraceutical also significantly reduced SUA concentration in participants with moderate and severe NAFLD compared with the placebo group.

A greater reduction of SUA concentration occurs in subjects with highest baseline UA values.

The results obtained in this secondary analysis are interesting because they encourage the use of natural molecules with nutraceutical effects for the management of metabolic diseases.

However, ad hoc studies are needed to evaluate the effects of this nutraceutical in hyperuricaemic individuals.

## 6. Patents

Bergacyn^®^ (provided by Herbal & Antioxidant SRL, Bianco, RC, Italy): one capsule containing a combination product containing bergamot polyphenolic fraction (BPF^®^), and wild type Cynara Cardunculus extract (CyC) plus excipients including PUFA and a mixture of bergamot pulp and albedo derivative]. (registered Patents RM2008A000615, PCT/IB2009/055061 and 102017000040866); (batch number 18R049, expiration date 10/2020).

## Figures and Tables

**Figure 1 medicina-58-01728-f001:**
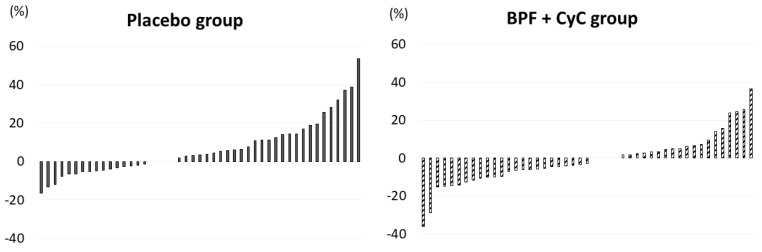
Individual reduction of SUA concentration according to treatment after 6 weeks. Abbreviation: BPF: polyphenolic fraction of bergamot; CyC: Cynara Cardunculus.

**Figure 2 medicina-58-01728-f002:**
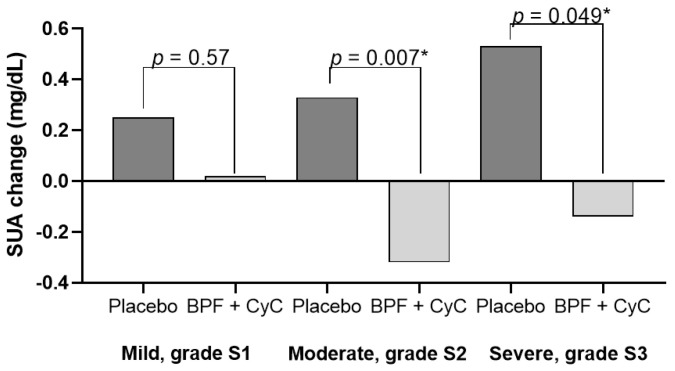
SUA change according to severity of hepatic steatosis at baseline. * Adjusted for age, gender, smoking habit, drugs, and weight change.

**Figure 3 medicina-58-01728-f003:**
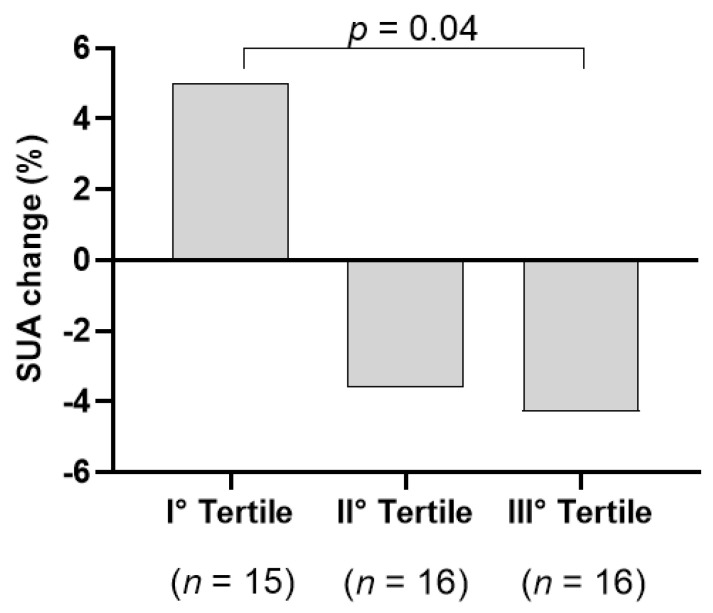
SUA change after 6 weeks according to tertiles of baseline SUA levels in individuals taking the nutraceutical.

**Table 1 medicina-58-01728-t001:** Baseline demographic and clinical characteristics of individuals with NAFLD according to the treatments.

Variables	Placebo (*n* = 47)	BPF + CyC (*n* = 47)	*p*-Value
Age (years)	50 ± 11	52 ± 9	0.37
Weight (Kg)	80.0 ± 10	79.9 ± 13	0.95
BMI (Kg/m^2^)	28.9 ± 4	29.1 ± 3	0.79
WHR	0.91 ± 0.1	1.12 ± 1.2	0.26
SBP (mmHg)	112 ± 17	112 ± 15	0.99
DBP (mmHg)	71 ± 14	73 ± 11	0.55
CAP score (dB/m)	286 ± 40	294 ± 39	0.30
IQR	11 ± 5	11 ± 4	0.96
Glucose (mg/dL)	92 ± 8	92 ± 7	0.75
HOMA-IR	2.28 ± 1.0	2.32 ± 1.3	0.88
TC (mg/dL)	193 ± 38	188 ± 34	0.49
TG (mg/dL)	115 ± 50	102 ± 46	0.16
HDL-C (mg/dL)	50 ± 12	53 ± 12	0.30
LDL-C (mg/dL)	120 ± 33	115 ± 31	0.46
AST (IU/L)	24 ± 15	21 ± 6	0.21
ALT (IU/L)	33 ± 28	24 ± 13	0.05
γGT (UI/L)	27 ± 20	25 ± 18	0.73
SUA (mg/dL)	5.2 ± 1.1	5.1 ± 1.1	0.82
Creatinine (mg/dL)	0.82 ± 0.1	0.81 ± 0.1	0.72
Total bilirubin (mg/dL)	0.62 ± 0.3	0.63 ± 0.3	0.95
**Prevalence**			
Gender (Male, %)	60	55	0.83
Physical activity (%)	47	49	1
Smokers (%)	9	28	0.030
Obesity (%)	40	32	0.87
Hypertension (%)	32	34	1
Hyperlipidemia (%)	52	46	0.67
Antihypertensive drugs (%)	26	30	0.81
Beta-blockers (%)	9	6	1
Diuretics (%)	6	11	0.71
Lipid-lowering agents (%)	9	17	0.35
Antiplatelet agents (%)	4	11	0.43
Urate-lowering drugs (%)	4	0	0.49

Note. BMI = body mass index, WHR = waist to hip ratio, SBP = systolic blood pressure, DBP = diastolic blood pressure, CAP = controlled attenuation parameter, IQR = interquartile range, HOMA-IR = homeostatic model assessment of insulin resistance, TC = total cholesterol, TG = triglycerides, HDL-C = high density lipoprotein cholesterol, LDL-C = low density lipoprotein cholesterol, AST = aspartate aminotransferase, ALT = alanine aminotransferase, γGT = gamma glutamyltransferase, SUA = serum uric acid.

**Table 2 medicina-58-01728-t002:** Changes in clinical parameters at follow-up according to the treatments after 6 weeks.

Variables	Placebo (*n* = 47)	BPF + CyC (*n* = 47)	*p*-Value
Weight (Kg)	−2.1 ± 1.8	−3.1 ± 2.3	0.031
BMI (Kg/m^2^)	−0.8 ± 0.6	−1.1 ± 0.8	0.026
WHR	−0.001 ± 0.04	−0.19 ± 1.2	0.29
Glucose (mg/dL)	−0.4 ± 6.7	−0.6 ± 6.7	0.87
HOMA-IR	−0.3 ± 0.8	−0.5 ± 1	0.50
TG (mg/dL)	−14.7 ± 39	0.6 ± 54	0.12
HDL-C (mg/dL)	−0.4 ± 6	−3 ± 5	0.34
LDL-C (mg/dL)	−2.6 ± 28	−9.1 ± 14	0.35
AST (IU/L)	−2.4 ± 6	−1.1 ± 6	0.34
ALT (IU/L)	−6.1 ± 13	−2.7 ± 6	0.10
γGT (UI/L)	−3.7 ± 9	−6.0 ± 9	0.22
SUA (mg/dL)	0.3 ± 0.7	−0.1 ± 0.7	0.004
aSUA (mg/dL) *	0.3 ± 0.1	−0.1 ± 0.1	0.005
SUA (%)	6.7 ± 14	−1.1 ± 13	0.008
aSUA (%) *	6.6 ± 2	−1.0 ± 2	0.012
Creatinine (mg/dL)	−0.004 ± 0.08	−0.006 ± 0.07	0.91

SUA adjusted for age, gender, smokers, sartans, calcium channel, β blockers, diuretics, urate-lowering drugs, antiplatelet agents, and weight change. Note. BMI = body mass index, WHR = waist to hip ratio, HOMA-IR = homeostatic model assessment of insulin resistance, TG = triglycerides, HDL-C = high density lipoprotein cholesterol, LDL-C = low density lipoprotein cholesterol, AST = aspartate aminotransferase, ALT = alanine aminotransferase, γGT = gamma glutamyltransferase, SUA = serum uric acid. * Adjusted for age, gender, smoking habit, drugs, and weight change.

**Table 3 medicina-58-01728-t003:** Multivariable linear regression analysis –Factors associated with absolute SUA change in individuals with Non-Alcoholic Fatty Liver Disease (NAFLD).

					C.I. 95%	
Dependent Variable SUA Change	B	SE	β	*p*-Value	LL	UL
Treatment	−0.43	0.14	−0.29	0.004	−0.71	−0.14

Note. Excluded variables: age, gender, smokers, and weight change. CI = confidence interval; LL = lower limit; UL = upper limit.

## Data Availability

The datasets generated for this study are available on request to the corresponding author.

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
