# Peer review of "Citrus Bergamia and Cynara Cardunculus Reduce Serum Uric Acid in Individuals with Non-Alcoholic Fatty Liver Disease"

_medicina, 2022, doi:10.3390/medicina58121728_

Round 1
Reviewer 1 Report
Summary
The authors studied the effect of the polyphenolic fraction of Citrus Bergamia in combination with the extract of Cynara Cardunculus (from now on “mix)” in NAFLD patients. Specifically, they evaluated the effect of the treatment on body weight and the serum levels of uric acid, which have been described to be increased in liver disease. The authors describe a significant reduction of uric acid in male subjects with fatty liver 6 weeks after starting the treatment, in comparison to the placebo group.
The study demonstrates a slight correlation between the reduction of SUA and the use of the mix in patients with NAFLD. However, these effects could be due to caloric restriction and weight loss. Transient elastography has acceptable diagnostic accuracy for advanced liver fibrosis and cirrhosis, but the authors use this diagnostic tool to detect hepatic steatosis. The authors include overweight/obese subjects in both groups, but the diagnostic tool (Fibroscan) could not be accurate in patients with high thoracic fat. Finally, the authors could explain better what is shown in the figures (%?), and the statistical tests employed.
Abstract
- Line 24: Easier to read: a greater reduction in serum uric acid reduction…
- Line 28 + 29: Abbreviations were not introduced previously (SUA, BPF, CyC)
Introduction
- Line 45: is 4.7-5.6 mg/mL the normal range? Please rephrase the sentence, and it is a bit misleading.
- Line 46: In line 44 they introduced to abbreviation CVD, then used CV disease later
- Line 50: typo error
Materials and methods
- 2.1. Study design: a group of patients treated with only BPF, and a group of patients treated with only CyC are missing to find out the effect of these separate compounds.
- 2.2. Population and randomization: the authors state that the study subjects were chosen because of the diagnosis of hepatic steatosis using transient elastography. This diagnostic tool is suitable for assessing liver fibrosis in patients with advanced liver disease, but not for diagnosing hepatic steatosis. Moreover, the authors do not separate the subjects into groups according to their level of severity of liver disease, or to the presence of liver fibrosis. Knowing how NAFLD level affects treatment efficacy would have been very interesting.
- The authors did not take into account the effect of the change in eating habits on the efficacy of the treatment.
- Line 75: Probable cause of bias -> People who don’t trust in plant-based drugs might not be included since they are less likely to sign up.
- Line 91: Why only include subjects from the first 6 weeks? -> They stated in the limitations that a longer period of time would be interesting (line 350)
Results
- Table 2: It is interesting, that both groups lost weight and lowered their BMI
- Figure 1: Is every bar representing an individual within the study?
Discussion
- The authors observed a significant effect of the treatment on male subjects but not on female subjects. It would be interesting for the authors to discuss why there is this difference in effect due to gender.
- Line 268: Why discuss the benefits of lowering SUA?
- Paragraph from line 283: Maybe in the introduction? -> Can explain in the introduction why Cynara and bergamia are used together with Line 316
- I am missing possible theories why both groups had reduced weight etc. and why there was a difference in men but not women
-
Figures
The figures in the manuscript are blurry.
Author Response
Review 1:
Summary
The authors studied the effect of the polyphenolic fraction of Citrus Bergamia in combination with the extract of Cynara Cardunculus (from now on “mix)” in NAFLD patients. Specifically, they evaluated the effect of the treatment on body weight and the serum levels of uric acid, which have been described to be increased in liver disease. The authors describe a significant reduction of uric acid in male subjects with fatty liver 6 weeks after starting the treatment, in comparison to the placebo group.
The study demonstrates a slight correlation between the reduction of SUA and the use of the mix in patients with NAFLD. However, these effects could be due to caloric restriction and weight loss. Transient elastography has acceptable diagnostic accuracy for advanced liver fibrosis and cirrhosis, but the authors use this diagnostic tool to detect hepatic steatosis. The authors include overweight/obese subjects in both groups, but the diagnostic tool (Fibroscan) could not be accurate in patients with high thoracic fat. Finally, the authors could explain better what is shown in the figures (%?), and the statistical tests employed.
Response: Many thanks for all these comments that improved our manuscript. We revised the statistical analysis and revised the summary of the manuscript as well as the diverse sections.
Abstract
- Line 24: Easier to read: a greater reduction in serum uric acid reduction…
Response: Many thanks for this suggestion. We changed this part.
- Line 28 + 29: Abbreviations were not introduced previously (SUA, BPF, CyC)
Response: We changed this part.
Introduction
- Line 45: is 4.7-5.6 mg/mL the normal range? Please rephrase the sentence, and it is a bit misleading.
Response: Many thanks for this suggestion we deleted the sentence.
- Line 46: In line 44 they introduced to abbreviation CVD, then used CV disease later
Response: we corrected.
- Line 50: typo error
Response: We corrected this typo.
Materials and methods
- 2.1. Study design: a group of patients treated with only BPF, and a group of patients treated with only CyC are missing to find out the effect of these separate compounds.
Response: Many thanks for this question. Clinical, preclinical and cellular studies have already been conducted to evaluate the effects of antioxidants from Bergamot alone on hepatic steatosis and on uric acid (Ferro et al.; Mirarchi et al.; Parafati et al.). A meta-analysis provides evidence from randomized trials to support the use of artichoke leaf extract as a hepatoprotective agent in patients with NAFLD (Kamel et al) and other studies focused on uric acid. So, we conducted a study to test the combined effect of Bergamot extract with artichoke. In the present work, we attribute the beneficial effects on uric acid and liver levels to the nutraceutical as a whole. We better explain the reasons in the introduction.
Ref. Ferro Y, Pujia R, Mazza E, Lascala L, Lodari O, Maurotti S, Pujia A, Montalcini T. A new nutraceutical (Livogen Plus®) im-proves liver steatosis in adults with non-alcoholic fatty liver disease. J Transl Med 2022, 20, 377.
Mirarchi A, Mare R, Musolino V, Nucera S, Mollace V, Pujia A, Montalcini T, Romeo S, Maurotti S. Bergamot Polyphenol Extract Reduces Hepatocyte Neutral Fat by Increasing Beta-Oxidation. Nutrients 2022,14-3434.
Parafati M, Lascala A, La Russa D, Mignogna C, Trimboli F, Morittu VM, Riillo C, Macirella R, Mollace V, Brunelli E, Janda E. Bergamot Polyphenols Boost Therapeutic Effects of the Diet on Non-Alcoholic Steatohepatitis (NASH) Induced by "Junk Food": Evidence for Anti-Inflammatory Activity. Nutrients 2018, 10-1604
Kamel, A. M., & Farag, M. A. Therapeutic Potential of Artichoke in the Treatment of Fatty Liver: A Systematic Review and Meta-Analysis. Journal of medicinal food, 2022, 25, 931–942.
- 2.2. Population and randomization: the authors state that the study subjects were chosen because of the diagnosis of hepatic steatosis using transient elastography. This diagnostic tool is suitable for assessing liver fibrosis in patients with advanced liver disease, but not for diagnosing hepatic steatosis. Moreover, the authors do not separate the subjects into groups according to their level of severity of liver disease, or to the presence of liver fibrosis. Knowing how NAFLD level affects treatment efficacy would have been very interesting.
Response: European guidelines for the management of NAFLD recommend, of course, using ultrasound (US) as first-choice imaging in adults at risk for NAFLD (EASL et al.). However, in clinical practice, it is well accepted that US limitations include suboptimal sensitivity and specificity for the detection of mild steatosis less than 25–30% of hepatocytes involved and operator dependence; furthermore, there are also concerns about the possible consequences of overdiagnosis of NAFLD in severe grades. Therefore, to date, it can be used to obtain only a qualitative and semiquantitative assessment of the extent of disease. Instead, magnetic resonance imaging, the ideal technique, with the highest sensitivity to detect liver steatosis, to date, it is not recommended as a first-line tool given its cost and limited availability in this clinical setting. CAP appears to be a new diagnostic tool for non-invasive assessment and quantification of steatosis, enhancing the spectrum of the non-invasive method for the exploration and follow-up of the patients with fatty liver. The performance of CAP was evaluated in the study of Sasso et al. taking the histological grade of steatosis as reference. AUROC was equal to 0.91 and 0.95 for the detection of more than 10 and 33% of steatosis, respectively. These data showed that CAP can efficiently separate several steatosis grades. A recent study focusing on an head to head comparison between US and TE (Salmi et al.), demonstrated that CAP values obtained with TE were comparable with US diagnosis and closely correlated with the steatosis severity of US detected in high-risk subjects. Moreover, in low-risk patients, CAP seems to be more sensitive in steatosis detection compared to US evaluation. The CAP evaluation by FibroScan has been thus suggested to be used to diagnose and detect, with over the 90% of sensitivity, liver steatosis in recent practice guidelines (de Barros et al.). The CAP value is used also in obese individuals as showed by De Barros et al. and the performance of TE was evaluated in obese taking the histological grade of steatosis as reference (Eilenberg et al.).
We added these points in the discussion section.
As, suggested, we added the statistical analysis on the effect of the nutraceutical according to the liver steatosis grade (see in the statistical analysis section, results and discussion sections).
Ref. European Association for the Study of the Liver (EASL), European Association for the Study of Diabetes (EASD), & European Association for the Study of Obesity (EASO) (2016). EASL-EASD-EASO Clinical Practice Guidelines for the management of non-alcoholic fatty liver disease. Journal of hepatology, 64(6), 1388–1402.
Sasso, M., Beaugrand, M., de Ledinghen, V., Douvin, C., Marcellin, P., Poupon, R., Sandrin, L., & Miette, V. (2010). Controlled attenuation parameter (CAP): a novel VCTE™ guided ultrasonic attenuation measurement for the evaluation of hepatic steatosis: preliminary study and validation in a cohort of patients with chronic liver disease from various causes. Ultrasound in medicine & biology, 36(11), 1825–1835.
Salmi, A., di Filippo, L., Ferrari, C., Frara, S., & Giustina, A. (2022). Ultrasound and FibroScan® Controlled Attenuation Parameter in patients with MAFLD: head to head comparison in assessing liver steatosis. Endocrine, 78(2), 262–269.
- European Association for the Study of the Liver. Electronic address: easloffice@easloffice.eu, Clinical Practice Guideline Panel, Chair:, EASL Governing Board representative:, & Panel members: (2021). EASL Clinical Practice Guidelines on non-invasive tests for evaluation of liver disease severity and prognosis - 2021 update. Journal of hepatology, 75(3), 659–689.
de Barros, F., & Fonseca, A. (2020). Bariatric surgery during the evolution of fatty liver-A randomized clinical trial comparing gastric bypass and sleeve gastrectomy based on transient elastography. Clinical obesity, 10(6), e12393.
Eilenberg, M., Munda, P., Stift, J., Langer, F. B., Prager, G., Trauner, M., & Staufer, K. (2021). Accuracy of non-invasive liver stiffness measurement and steatosis quantification in patients with severe and morbid obesity. Hepatobiliary surgery and nutrition, 10(5), 610–622.
- The authors did not take into account the effect of the change in eating habits on the efficacy of the treatment.
Response: In this study, we provided only general verbal dietary advice to both groups to adhere to a Mediterranean diet pattern. However, in this revised version of the manuscript, we categorized the population in tertiles of SUA and we find a significant reduction in SUA only in those with a high basal SUA level (III Tertile). Tertiles were comparable for body weight change (among tertiles p= 0.12). We therefore hypothesize an effect on uric acid that is independent of weight loss and change in eating habits. We added this finding in the discussion.
- Line 75: Probable cause of bias -> People who don’t trust in plant-based drugs might not be included since they are less likely to sign up.
Response: We agree. However, the enrollment in trials is always controversial. It has been suggested that medical/pharmacological trials are subject to selection bias: experiments on a new treatment attract patients that are optimistic about the new treatment, and these patients probably respond better to that treatment. In drug studies, those who are afraid of taking drugs may be excluded. Furthermore, it is more likely that subjects in good health are enrolled while those suffering from diseases refuse because they have difficulty coming to the facility. In prospective cohort studies enroll subjects who have not yet developed the health outcomes of interest, they cannot have selection bias as a result of enrollment procedures, but they can have selection bias as a result of differential retention during follow up.
These concerns is therefore present in all trials and not only in ours.
Ref. Malani, A. Patient enrollment in medical trials: Selection bias in a randomized experiment. Journal of Econometrics, 2008, 144, 341-351.
- Line 91: Why only include subjects from the first 6 weeks? -> They stated in the limitations that a longer period of time would be interesting (line 350)
Response: The intervention in the present study was designed to provide proof of principle and only one dose was tested. It would be of interest to enroll individuals with hyperuricemia in a future clinical trial with a longer study duration. We added this point to the limitations
Results
- Table 2: It is interesting, that both groups lost weight and lowered their BMI
Response: This phenomenon is well described in the literature (Bouzalmate Hajjaj et al.). In fact, we hypothesize an effect on uric acid that is independent of weight loss. When we categorized the population in SUA tertiles, we find the greatest reduction in SUA levels in those with a high basal SUA level ( III Tertile) and the tertiles were comparable for body weight change (among tertiles p= 0.12). This may confirm that the effect of nutraceutical on uric acid is independent of weight loss.
Ref. Bouzalmate Hajjaj, A., Massó Guijarro, P., Khan, K. S., Bueno-Cavanillas, A., & Cano-Ibáñez, N. A systematic review and meta-analysis of weight loss in control group participants of lifestyle randomized trials. Scientific reports, 2022, 12, 12252.
- Figure 1: Is every bar representing an individual within the study?
Response: Yes, we confirm it.
Discussion
- The authors observed a significant effect of the treatment on male subjects but not on female subjects. It would be interesting for the authors to discuss why there is this difference in effect due to gender.
Response: In this revised version, we find a significant reduction in uric acid in all those with high basal acid uric level (>5.4 mg/dL, III tertile, both genders) (see figure 2). In this study, we unfortunately enrolled less female than men (60% males). Among participants in the III Tertile only 19%were females. We cannot therefore exclude the effect also on women. We added all these new findings in the new version of the manuscript.
- Line 268: Why discuss the benefits of lowering SUA?
Response: We deleted the most part of this point.
- Paragraph from line 283: Maybe in the introduction? Can explain in the introduction why Cynara and bergamia are used together with Line 316
Response: We changed the text according to this suggestion. As suggested we explained in the introduction this point.
- I am missing possible theories why both groups had reduced weight etc. and why there was a difference in men but not women
Response: We agree with the reviewer and revised all the statistical analysis and rewrote manuscript. In the whole population, we performed GLM test to adjust SUA change also for weight change, confirming the higher reduction in BPF+CyC group than in placebo group. Futhermore, when we categorized the population in SUA tertiles, that were comparable for body weight change, we find the greatest reduction in SUA levels in those with a high basal SUA level. This may confirm that the effect of BPF+CyC on uric acid is independent of weight loss.
Figures
The figures in the manuscript are blurry.
Response: We have improved the quality of the figures.

Reviewer 2 Report
In the presented article, the authors discussed the topic of the Use of Citrus Bergamia and Cynara Cardunculus for reduce serum uric acid in individuals with non-alcoholic fatty liver disease.
As is well known, inflammation generates large amounts of free radicals. Substances with a potential antioxidant effect may mitigate the harmful effects of free radicals on the body.
In their research, the authors proposed the use of Citrus Bergamia and Cynara Cardunculus extract as antioxidant-containing products in patients with fatty liver disease and elevated uric acid levels. The research presented by the authors is very interesting and applicable.
The authors used a large research group in their research. The summary work is very good. The results were broadly discussed. This work has very practical applications. I propose to accept this quality paper and publish it in the Medicina.
Author Response
Response. Many thanks for these comments.
Round 2
Reviewer 1 Report
The authors have addressed all the comments, and the manuscript has been improved after the revision. The reviewer recommends the acceptance of the study.